# Cytogenetic Assessment and Risk Stratification in Myelofibrosis with Optical Genome Mapping

**DOI:** 10.3390/cancers15113039

**Published:** 2023-06-02

**Authors:** Álvaro Díaz-González, Elvira Mora, Gayane Avetisyan, Santiago Furió, Rosalía De la Puerta, José Vicente Gil, Alessandro Liquori, Eva Villamón, Carmen García-Hernández, Marta Santiago, Cristian García-Ruiz, Marta Llop, Blanca Ferrer-Lores, Eva Barragán, Silvia García-Palomares, Empar Mayordomo, Irene Luna, Ana Vicente, Lourdes Cordón, Leonor Senent, Alberto Álvarez-Larrán, José Cervera, Javier De la Rubia, Juan Carlos Hernández-Boluda, Esperanza Such

**Affiliations:** 1Hematology Research Group, Instituto de Investigación Sanitaria La Fe, 46026 Valencia, Spain; diaz_alv@gva.es (Á.D.-G.);; 2Department of Hematology, Hospital Universitario y Politécnico La Fe, 46026 Valencia, Spain; 3Department of Hematology, Hospital Arnau de Vilanova, 46015 Valencia, Spain; 4Centro de Investigación Biomédica en Red de Cáncer (CIBERONC), 28029 Madrid, Spain; 5Department of Hematology, Hospital Clínico Universitario–INCLIVA, 46010 Valencia, Spain; 6Department of Hematology, Hospital General Universitario de Alicante, 03010 Alicante, Spain; 7Molecular Biology Unit, Clinical Analysis Service, Hospital Universitario y Politécnico La Fe, 46026 Valencia, Spain; 8Pathology Department, Hospital Universitario y Politécnico La Fe, 46026 Valencia, Spain; 9Department of Hematology, Hospital Clínic, 08036 Barcelona, Spain; 10Genetics Department, Hospital Universitario y Politécnico La Fe, 46026 Valencia, Spain; 11School of Medicine and Dentistry, Catholic University of Valencia, 46001 Valencia, Spain

**Keywords:** myelofibrosis, optical genome mapping, chromosome banding analysis, prognosis

## Abstract

**Simple Summary:**

Cytogenetic risk categorization is essential for the application of prognostic scores and the management of patients with myelofibrosis (MF). However, the low resolution of conventional karyotyping and the absence of metaphases are major limitations in MF. Herein, cytogenetic characterization via optical genome mapping (OGM) was performed in a cohort of 21 MF patients. To evaluate OGM impact on prognosis assessment, risk stratification scores were recalculated, including all the cytogenetic alterations uncovered. OGM satisfactorily characterized all patients with a previously unsuccessful karyotype. Additionally, the cryptic alterations detected via OGM led to an upgrade in the risk category for three patients. In conclusion, OGM may be a promising technique for cytogenetic assessment in MF.

**Abstract:**

Cytogenetic assessment in myelofibrosis is essential for risk stratification and patient management. However, an informative karyotype is unavailable in a significant proportion of patients. Optical genome mapping (OGM) is a promising technique that allows for a high-resolution assessment of chromosomal aberrations (structural variants, copy number variants, and loss of heterozygosity) in a single workflow. In this study, peripheral blood samples from a series of 21 myelofibrosis patients were analyzed via OGM. We assessed the clinical impact of the application of OGM for disease risk stratification using the DIPSS-plus, GIPSS, and MIPSS70+v2 prognostic scores compared with the standard-of-care approach. OGM, in combination with NGS, allowed for risk classification in all cases, compared to only 52% when conventional techniques were used. Cases with unsuccessful karyotypes (*n* = 10) using conventional techniques were fully characterized using OGM. In total, 19 additional cryptic aberrations were identified in 9 out of 21 patients (43%). No alterations were found via OGM in 4/21 patients with previously normal karyotypes. OGM upgraded the risk category for three patients with available karyotypes. This is the first study using OGM in myelofibrosis. Our data support that OGM is a valuable tool that can greatly contribute to improve disease risk stratification in myelofibrosis patients.

## 1. Introduction

Myelofibrosis (MF) is a *BCR::ABL1*-negative chronic myeloproliferative neoplasm characterized by the presence of “driver” mutations in *JAK2*, *CALR*, and *MPL* genes [1]. MF can present de novo (Primary MF, PMF) [2] or as secondary MF (SMF) due to the evolution of polycythemia vera (PPV-MF) or essential thrombocythemia (PET-MF), but both presentations have similar clinical and biological characteristics [3].

Hematopoietic Stem Cell Transplantation (HSCT) is the only curative treatment in MF [4], but it is associated with a high rate of transplant-related mortality and severe morbidity [5]. Consequently, in whom and when HSCT is performed should be carefully considered. HSCT is usually indicated in high-risk patients, as defined by prognostic models, so achieving a complete characterization of the disease allows for a tailored treatment [6,7]. In the last decade, different studies have highlighted that karyotype abnormalities have a strong impact on MF prognosis. Therefore, cytogenetics is currently integrated into MF prognostic models used in routine clinical practice such as the Dynamic International Prognostic Scoring System (DIPSS-plus), Genetically Inspired Prognostic Scoring System (GIPSS), or Mutation-enhanced International Prognostic Score System for Transplantation-Age Patients with Primary Myelofibrosis version 2.0 (MIPSS70+v2) [8,9,10]. Unfortunately, the karyotype is not available in approximately 50% of MF cases in the multicenter real-world series due to bone marrow fibrosis, poor in vitro cell growth and other disease-related factors [11,12,13]. Fluorescent in situ hybridization (FISH) and chromosomal microarrays (CMAs) can be used to overcome these limitations. Nevertheless, FISH has a limited utility due to a probe-dependent approach [14], and CMAs cannot detect balanced events such as translocations or inversions [15].

Optical genome mapping (OGM) is a novel high-throughput diagnostic technique that may overcome the drawbacks of standard cytogenetic techniques. OGM is capable of detecting structural variants (SVs) from 500 base pairs in a single assay and with higher sensitivity than conventional techniques [16]. This technology has already been applied to other hematological neoplasms such as leukemia [17,18,19,20], myelodysplastic syndromes [21], and chronic lymphocytic leukemia [22] with promising results.

The aim of this study was to assess the application of OGM in patients with MF. The results were compared to karyotype to evaluate the impact on prognostic cytogenetic reclassification. To our knowledge, this is the first study that analyzes the performance of OGM in MF. Given the frequently encountered difficulties for karyotyping in MF, we believe that our results can refine the current risk stratification and improve the outcome of these patients. Additionally, the identification of novel chromosomal abnormalities may further increase our understanding of the disease.

## 2. Materials and Methods

### 2.1. Sample Selection

Patients who fulfilled the PMF World Health Organization (WHO)-defined criteria [2] and the PPV-MF and PET-MF International Working Group-Myeloproliferative Neoplasm Research and Treatment (IWG-MRT) criteria [23] were included in the study. A total of 21 MF patients (12 PMF, 2 PPV-MF, and 7 PET-MF) from three different institutions (Hospital La Fe *n* = 19, Hospital Clínico de Valencia *n* = 1 and Hospital General Alicante *n* = 1) were selected. Samples were obtained prospectively. Peripheral blood (PB) with ethylenediaminetetraacetic acid (EDTA) was frozen at −80 °C on arrival at the Biobank of our institution following the manufacturer’s recommendations (Bionano Genomics, San Diego, CA, USA). All samples were stored until the day of processing. The study was conducted in compliance with the principles of the Declaration of Helsinki, Good Clinical Practice, and all applicable regulatory requirements. Approval by the Ethical Committee of the La Fe Health Research Institute was obtained.

### 2.2. Conventional Testing

All cases were analyzed following the conventional workflow, which includes chromosome banding analysis (CBA) and Next-Generation Sequencing (NGS). CBA was performed from bone marrow following standard procedures. Chromosomes were stained using G-banding, and karyotypes were reported according to the International System for Human Cytogenomic Nomenclature (ISCN, 2020 recommendations) [24]. NGS libraries were prepared using the SOPHiA Myeloid Solution™ kit (SOPHiA GENETICS, Rolle, Switzerland). This commercial panel can detect single nucleotide variants (SNVs) and CNVs of 30 genes related to myeloproliferative neoplasms and leukemia: *ABL1, ASXL1, BRAF, CALR, CBL, CEBPA, CSF3R, DNMT3A, ETV6, EZH2, FLT3, HRAS, IDH1, IDH2, JAK2, KIT, KRAS, MPL, NPM1, NRAS, PTPN11, RUNX1, SETBP1, SF3B1, SRSF2, TET2, TP53, U2AF1, WT1*, and *ZRSR2*.

### 2.3. Optical Genome Mapping

Samples frozen at −80 °C were thawed in a 37 °C water bath for 2 min. Ultra-high-molecular weight (UHMW) DNA was extracted following the manufacturer’s instructions (Bionano Genomics, San Diego, CA, USA). After counting the white blood cells (WBCs) in each sample, a spin was performed (2200× *g*) to obtain a pellet, and the supernatant was removed. Afterwards, the WBCs were lysed and digested with proteinase K and treated with phenylmethylsulphonyl fluoride solution (PMSF) to obtain genomic DNA (gDNA). A paramagnetic nanobind disk was added to bind the gDNA to the disk, and then gDNA wash and elution were performed. The gDNA obtained was kept at room temperature overnight for homogenization. On the following day, gDNA quantitation was performed using a Qubit™ dsDNA BR Assay Kit with a Qubit 3.0 Fluorometer (ThermoFisher Scientific, Waltham, MA, USA) to check the quality of the gDNA and the homogenization of the sample (defined by a coefficient of variation less than 0.3). In total, 750 nanograms of DNA were labeled with a DLE-1 enzyme which places 500,000 fluorescent labels throughout the genome at a specific sequence motif (CTTAAG) occurring approximately 15 times per 100 kb. The UHMW DNA label was kept at room temperature overnight. On the next day, gDNA quantitation was repeated using a Qubit™ dsDNA HR Assay to check that samples had sufficient quality to be analyzed. Labeled DNA was linearized in nanochannel arrays on a Saphyr chip^®^ and imaged with an extremely high-throughput and in an automated manner using the Saphyr^®^ Instrument. In total, 3 samples were loaded per chip, with a maximum of 2 chips per run and a runtime limit of 65 h. Analysis of the samples was performed with the Bionano Access software. Changes in the patterning or spacing of the labels were detected automatically and compared to a reference genome (hg38) to call genome-wide structural variants. For each sample, the objective was to collect 1500 Gbps of data to achieve an effective coverage of 300x, which reaches 90% sensitivity when detecting structural variants at a 5% variant allele fraction (VAF), as described in the manufacturer’s protocol.

### 2.4. Data Analysis Strategy

Analysis of the samples was performed with the Bionano Access software. The results were analyzed with two pipelines: de novo assembly for detection of SVs and copy neutral loss of heterozygosity (CN-LOH), and a rare variant pipeline (RVP) for analysis of SVs at low allelic fractions. The de novo assembly pipeline detects germline SVs by building consensus maps and comparing these maps to a reference genome (GRCh38/hg38). This pipeline can detect homozygous and heterozygous SVs ≥500 bp based on the differences in the alignment of labels between the sample assembly and the reference assembly. In contrast, the RVP allows for SV detection by looking for mismatches between molecules and the reference genome (GRCh38/hg38) by comparing the molecules directly to the reference. This allows the pipeline to detect SVs ≥5 kbp and down to 5% allele fractions with adequate coverage.

Default confidence scores (range 0–1) for SVs were used for the RVP and de novo assembly pipelines: 0 for insertions, 0 for deletions, 0.7 for inversions, and 0.05 for intra-fusion and inter-translocation. The confidence score for duplications was undefined and set to −1 according to the manufacturer’s recommendations. After applying the described filters, significant SVs bigger than 500 bp were reported. For CNVs, the recommended filters were also used, which include a confidence score of 0.99 for a minimum size of 500 kbp. CN-LOH detection required a minimum size of 25 Mb according to the recommended filters. Each alteration was compared to Bionano’s human control sample SV database, which contains variants collected from ethnically diverse mapped human genomes with no reported disease phenotypes. To filter the variants, we first selected those that were present in less than 1% of the population database. Secondly, to exclude variants located in high-variance regions, we evaluated only structural variations (SVs) and copy number abnormalities in non-masked regions. Thirdly, we reported variants with a VAF > 5% and that were present in at least 5 molecules. Finally, we manually inspected each alteration with at least two reviewers to identify true calls. These calls were classified as Tier 1 to Tier 3 and validated using an alternative technique.

### 2.5. Validation of the Results

To validate the novel alterations detected via OGM, we accomplished two strategies. Firstly, FISH was performed with specific probes to confirm SVs in those cases with translocations. In addition, to validate CNVs and CN-LOH, a CMA was performed with Cytoscan HD according to the manufacturer’s protocol (Affymetrix, Santa Clara, CA, USA). Data were analyzed by using Chromosome Analysis Suite (ChAS) software 4.2.0.80 version (Affymetrix). Filters applied for the detection of CNAs were ≥10 consecutive markers in a region of at least 10 kb, and ≥200 markers in at least 10 Mb for regions of CN-LOH. The human reference GRCh38/hg38 assembly was used for alignment.

## 3. Results

### 3.1. Baseline Characteristics

Baseline patient characteristics at the time of obtaining the sample for OGM are shown in Table 1. CBA was informative in 11/21 (52%) patients, being normal in 7 patients and abnormal in 4 and non-informative in the remaining 10/21 (48%). NGS was conducted for all patients. The distribution of driver mutations was: 58% (12/21) *JAK2*, 34% (7/21) *CALR*, 4% (1/21) *MPL*, and 4% (1/21) triple negative. A prognostically favorable type 1-like *CALR* mutation was detected in six of the seven patients with mutated *CALR*. Eleven patients (52%) presented one or more high molecular risk mutations in *ASXL1*, *SRSF2*, or *U2AF1*, as defined using the GIPSS. Additionally, two of these patients had mutations in *EZH2*. No *IDH1/2* mutations were detected.

### 3.2. Optical Genome Mapping Results

All samples processed met the quality criteria previously described [25]. A median label density of 15.66/100 kpb, a map rate of 87.2%, and an effective coverage of 425× was obtained. DNA quality and metrics for each run are detailed in Appendix A. Sample processing, chip run, and analysis required an average of 7 days. Overall, after applying the recommended filters, OGM detected a mean of 27 SVs per patient (range 16–48). CNVs were solely detected in 7 patients (range 1–6). In total, 8 SVs, 17 CNVs, and 1 CN-LOH were considered true calls. The OGM findings in comparison to standard techniques are shown in Table 2. Figure 1 displays the OGM data output in selected cases.

Cases with unsuccessful karyotypes (*n* = 10) via CBA could be fully assessed via OGM. A normal karyotype was identified in eight patients, and abnormal karyotypes were noted in two. Patient #5 showed a loss on the long arm of chromosome 7, a gain on the short arm of chromosome 9, and a gain on the long arm of chromosome 1: ogm[GRCh38] 1q21.1q44 × 3, 7q11.21q36.3x1, (9)x3. NGS analysis previously performed could infer the existence of CNVs on chromosome 7 (*EZH2* and *BRAF* x1.5 copies) and chromosome 9 (*ABL1* and *JAK2* x2.6 copies). Patient #21 carried two translocations, t(1;12)(p35.2;q13.13) and t(1;14)(p35.2;q32.31), that did not generate any fusion gene.

OGM was able to identify the majority of cytogenetic alterations reported by banding cytogenetics and confirmed normal karyotype in 4/11 cases (36%). Furthermore, OGM also detected additional cryptic aberrations not identified with standard techniques in 7 out of 11 patients with informative karyotypes. Cases #6, #11, and #19 presented cytogenetically missed deletions. Patient #6 displayed a 376 kbp deletion on the long arm of chromosome 4 (region 4q24), originating a homozygous deletion of *TET2*. In patients #11 and #19, deletions on chromosome 20 (20q11.21q11.22 and 20q11.21q13.32, respectively) were found, with sizes of 2.3 Mb and 26.8 Mb. All deletions resulted in a deletion of *ASXL1*. Furthermore, OGM showed additional rearrangement complexity in patient #10 with translocations t(2;11)(q37.1;q23.2) and t(7;11)(q31.31;q24.1) and in patient #16 with the translocation t(2;14)(p23.2;q32.12), accompanied by deletions flanking the translocation breakpoints in both patients. OGM also allowed for a more precise mapping of t(7;13)(q34;q14.2) in patient #11. In patient #4, CBA demonstrated a deletion on the long arm of chromosome X and a translocation leading to a derivative chromosome, der(6)t(1;6)(q10;p10). The resulting net imbalance of this abnormality was a trisomy of the long arm of chromosome 1 and a monosomy of the short arm of chromosome 6. OGM was able to detect a gain of material on the long arm of chromosome 1 and a loss of material on the short arm of chromosome 6. However, OGM did not identify the translocation t(1;6) due to the inability to accurately map centromeric regions. In addition, OGM discovered in this case a novel translocation t(12;17)(q24.31;p13.1), resulting in gene fusion *KDM2B::TP53.* De novo analysis did not detect significant additional SVs previously identified using a RVP (see Appendix A). However, de novo assembly was unable to detect the deletion in 4q24 of patient #6 (VAF 6%) and the translocations t(1;12)(p35.2;q13.13) and t(1;14)(p35.2;q32.31) in patient #21 (VAF 5% and 14%, respectively). This may be attributed to the fact that de novo assembly is designed for constitutional samples, with a lower limit of detection of approximately 25% VAF for events. Nevertheless, an LOH in the short arm of chromosome 9 affecting the *JAK2* gene in patient #9 could be identified solely via the de novo strategy.

Ultimately, OGM identified 19 additional cryptic aberrations in 9 out of 21 patients (43%). All the SVs as well as the CN-LOH of patient #9 were confirmed via FISH or CMA (see Appendix A). All novel findings produced via OGM were validated except in two cases (patient #16 and patient #21). Although FISH did not detect the translocation t(2;14) of patient #16, a microarray confirmed the deletions in the regions adjacent to the translocation. It was detected with the recommended filters and with a confidence score of 0.99 and a variant allele frequency (VAF) of 42%, which suggests it escaped detection from conventional techniques (see Appendix A). Confirmation of translocations t(1;12) and t(1;14) found in patient #21 was not possible due to the unavailability of a bone marrow sample.

### 3.3. OGM Impact on Risk Stratification

To evaluate the clinical impact on risk stratification, DIPSS-Plus, GIPSS, and MIPSSv2 prognostic scores were calculated using the baseline characteristics, NGS results, and cytogenetic results obtained through CBA. Only patients with initially informative CBA were considered when it came to applying the prognostic scale, as unsuccessful CBA does not allow for the complete classification of the patient. Based on these results, 4 out of 21 patients (#4, #10, #11, and #16) were initially classified as “high risk”. Subsequently, the CBA result was replaced with the new cytogenetic information obtained via OGM and the scores were recalculated, as in Table 3 and Figure 2. Patients older than 70-years-old (*n* = 8) were excluded for the calculation of MIPSS70+v2, as this prognostic scale was developed for patients aged 70-years-old or younger. The combination of OGM with NGS and baseline data enabled for the risk classification of all patients, in contrast to only 11 (52%) patients using the standard approach. OGM upgraded the prognostic scores in three patients (#6, #16, and #19) with a previously successful karyotype via CBA. No patient was downgraded after OGM analysis. In patient #10, OGM findings (11q deletion) increased the cytogenetic risk by going from an “unfavorable” to “high risk” karyotype. However, the prognostic scores remained unchanged as they had been already classified as “very-high risk” based on the CBA results, NGS, and baseline characteristics. Cytogenetic alterations of clinical unknown significance using current prognostic scores that did not modify the prognostic score were identified in patients #4 (*KMD2B::TP53* fusion gene) and #9 (CN-LOH in 9p).

Furthermore, OGM allowed for risk stratification in 10 patients with unsuccessful karyotypes. In 8 out of 10 patients, no significant alterations were found through OGM. Thus, these patients were classified into their respective risk categories primarily based on baseline data and high-risk mutations initially detected via NGS. In fact, four out of eight patients were reclassified as “high risk” (#1, #2, #7, and #20), despite having a normal chromosomal formula. For these patients, the primary benefit of OGM was to facilitate a cytogenetic study, ensuring the availability of all variables for patient classification. We identified high-risk aberrations with an impact on prognostic scores in the remaining two patients with an unsuccessful karyotype. Patient #5 showed a 7q deletion recognized as “high risk” for DIPSS-PLUS and “unfavorable” for GIPSS and MIPPSv70+2. Patient #21, with the two novel observed translocations t(1;12)(p35.2;q13.13) and t(1;14)(p35.2;q32.31), was cytogenetically classified as “unfavorable” for GIPSS and MIPSS70+v2. Nevertheless, these translocations did not meet the criteria to be classified as “unfavorable” in DIPSS-PLUS.

## 4. Discussion

In this study, we demonstrate that OGM is a promising alternative to banding cytogenetics in patients with MF. OGM enabled for risk stratification in all cases, compared to 61% when applying standard-of-care techniques. Moreover, this technique identified in peripheral blood the vast majority of chromosomal alterations detected via standard methods and additional cryptic aberrations included in prognostic scoring systems that could imply changes in therapeutic approach. Therefore, the OGM strategy could overcome the limitations of cytogenetic assessment in MF.

The incorporation of emerging techniques is essential to improve cytogenetic assessment in MF. The absence of dividing myeloid cells in peripheral blood may limit the feasibility of performing a conventional cytogenetic analysis (CBA) in PB. Therefore, BM CBA is the gold standard technique for cytogenetic evaluation [14]. Nevertheless, a high ratio of unsuccessful BM CBA is frequently encountered due to intrinsic disease-related factors. In fact, in this work, the frequency of failed CBA from bone marrow samples was 48%. On the other hand, to perform a multiprobe FISH panel covering all the SVs with a prognostic impact on MF is a laborious, time-consuming, and expensive strategy, not feasible in routine clinical practice. Furthermore, for appropriate classification according to the revised cytogenetics classification in primary myelofibrosis [26], it is necessary to know if an abnormality is the only one present, and this information cannot be obtained solely through FISH analysis. According to previous studies, FISH analysis in peripheral blood is comparable to the karyotype in bone marrow [27]. OGM is a genome-wide approach for SV evaluation in non-dividing cells that can be performed either in BM or PB, so we hypothesize conducting a study on PB using OGM. A total of 21 MF cases were analyzed, and OGM disclosed the alterations previously identified using standard testing and enabled cytogenetic evaluation in cases with non-informative karyotypes. Thus, performing a cytogenetic study via OGM in PB is a suitable alternative for these patients.

The novel alterations detected in cases with informative karyotypes allowed for risk category upgrade in three patients. Other authors have also reported OGM’s impact on the risk stratification of patients with myelodysplastic syndromes according to the IPSS-R [28]. “High-risk” and “favorable” cytogenetic alterations are well-defined in MF prognostic scales. Chromosomal aberrations not included in these categories are frequently classified as “unfavorable” [8,9]. Therefore, the higher resolution of OGM may lead to overdiagnosis of chromosomal alterations and overestimation of cytogenetic risk. Thus, it is necessary to be cautious, and only thoroughly validated alterations should be reported. Future studies in large patient samples are needed to better define prognostic associations of the abnormalities detected via OGM and other high-resolution techniques. In addition, OGM can detect rearrangements with functional deleterious effects. In our series, we disclosed one potentially disruptive translocation, t(12;17)(q24.31;p13.1), which involved *KDM2B* and *TP53* genes in patient #4. Increased *KMD2B* expression could facilitate the proliferation of hematopoietic progenitor cells and induced leukemic transformation according to a previous study [29], and the *TP53* gene is a well-recognized prognostic predictor in several hematological malignancies [30]. *TP53* mutations are rare (4%) in PMF patients and are not included in current prognostic scales [31]. However, *TP53* mutations are more common in advanced MF and are related to a high risk of transformation to acute myeloid leukemia and subsequent early death [32]. *TP53* rearrangements are rare, with only a few cases described in pediatric patients with osteosarcoma [33]. To date, no *KDM2B::TP53* rearrangements have been described, so this translocation should be seen as a potentially disrupting *TP53* function. Functional studies are needed to elucidate the role of all potential fusion genes reported via OGM analysis.

Similar to what was reported for myelodysplastic syndromes [28], our study demonstrated that both rare variant and de novo pipeline strategies are interesting approaches. The de novo assembly pipeline confirmed all the CNVs detected with the rare variant pipeline and, additionally, it was able to detect a CN-LOH affecting, among others, *JAK2* in the short arm of chromosome 9, which was not identified via rare variant analysis. Interestingly, the combination of the *JAK2* mutation and chromosome 9p LOH was recognized as a distinct genomic subgroup among myeloproliferative neoplasms [32]. This group was characterized by a higher proportion of polycythemia vera and also a higher risk of progression to secondary myelofibrosis. Actually, our patient with a 9p CN-LOH had a PPV-MF. LOH affecting chromosomes 17 or 20 that defined other genomic subgroups was not identified in this study. Furthermore, *TP53* CN-LOH was recently added as a predictor of adverse outcomes in the recent Molecular International Prognostic Scoring System for Myelodysplastic Syndromes [34]. Consequently, the widespread application of OGM in myeloproliferative neoplasm may contribute to identifying novel CN-LOH with prognostic impact.

However, OGM has limitations. Firstly, the strategy for obtaining UHMW DNA and labeling is manual, which requires a long processing time (median of 7 days in our study). In addition, careful handling of the sample when pipetting is critical for DNA quality. Consequently, clinicians must consider the total sample processing and analysis time in cases that require a rapid diagnosis. Secondly, OGM does not accurately map repetitive and centromeric regions of the genome. Therefore, this technique does not provide the detailed individual structure of each chromosome as observed in a karyotype. However, OGM can identify the rearrangements that often occur in euchromatic sequences at dicentric chromosome breakpoints. Additionally, although OGM cannot visualize the structure of an isochromosome, the imbalance in copy number is still observable. In fact, OGM missed the derivative chromosome der(6)t(1;6)(q10;p10) detected via CBA in patient #4. Thirdly, the presence of abnormal myelofibrosis cells intermingled with normal non-cancerous cells occurs in a mosaic pattern. While CBA provides insights into clonal architecture at the single-cell level, OGM examines the DNA of all cells. The accuracy of detecting subclones through OGM is dependent on the cut-off values used in the analysis pipelines. It is important to consider this to minimize the risk of failing to detect subclonal alterations. Finally, the t(2;14) in patient #16 was not detected via CBA, but OGM identified it at 42% VAF and CMA showed copy losses near the breakpoints, providing evidence of the translocation’s existence. This translocation may have been cryptic to CBA due to the distal location of both breakpoints, as the resolution of CBA for MF is typically around 300–350 bands [14]. The loss of the entire 2p distal to 2p23.2 is approximately 28MB, and if replaced with ~12MB of chr14, the total displaced sequence would be 16MB, which is at the limit of detection for G-banding at 350 bands. To confirm this translocation, a metaphase FISH was performed using whole chromosomal painting since there were no specific probes covering the affected region. However, the metaphase FISH was unable to confirm the translocation, possibly due to metaphase bias and the growth advantage of the 46,XY cultured cells over the t(2;14) cells. Further studies are needed to better ascertain the methodological issues that may explain discrepancies between banding cytogenetics and OGM.

## 5. Conclusions

In summary, the application of OGM in this preliminary series of MF patients demonstrates its feasibility and impact on cytogenetic assessment and risk stratification in combination with NGS. Conventional cytogenetic techniques offer a limited resolution, whereas OGM is a powerful technique to disentangle the genomic complexity of MF. Future studies comprising a larger cohort of patients are necessary to better assess genomic alterations that contribute to MF prognosis and imply more intensive management. Thus, although further evidence is needed, our data suggest that OGM may help inform treatment decision-making in patients with MF.

## Figures and Tables

**Figure 1 cancers-15-03039-f001:**
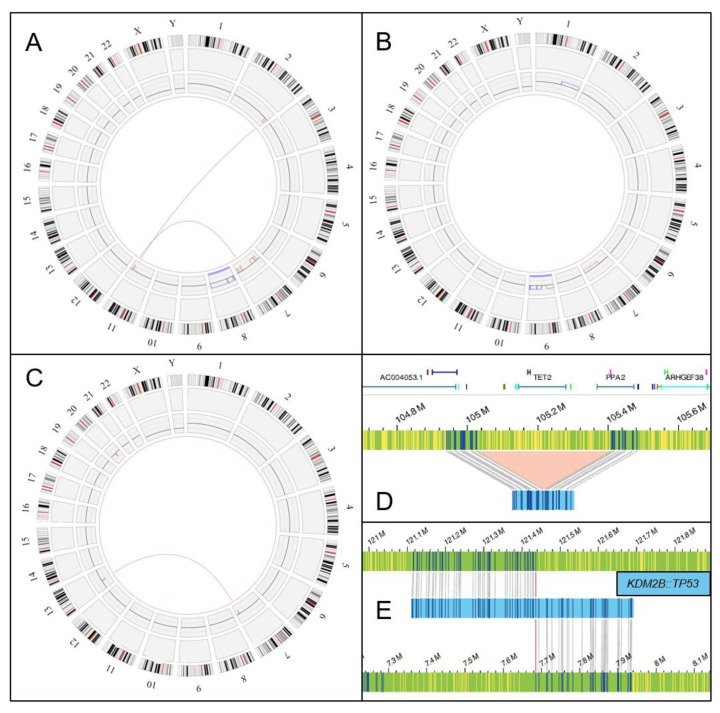
Optical genome mapping selected cases: circos plot and SV analysis. (**A**) Circos plot showing a complex event with t(2;11)(q37.1;q23.2) and t(7;11)(q31.31;q24.1) and with material losses in adjacent regions with a trisomy of chromosome 8 (patient #10); (**B**) gain of material in region 1q21.1q44, loss of material in region 7q11.21q36.3, and trisomy of chromosome 9 (patient #5); (**C**) translocation t(7;13)(q34;q14.2) with material losses in adjacent regions and deletion of 20q11.21q11.22 (patient #11); (**D**) deletion in region 4q24 involving a homozygous deletion of *TET2* (patient #6); (**E**) *KDM2B::TP53* gene fusion resulting from t(12;17)(q24.31;p13.1) (patient #4).

**Figure 2 cancers-15-03039-f002:**
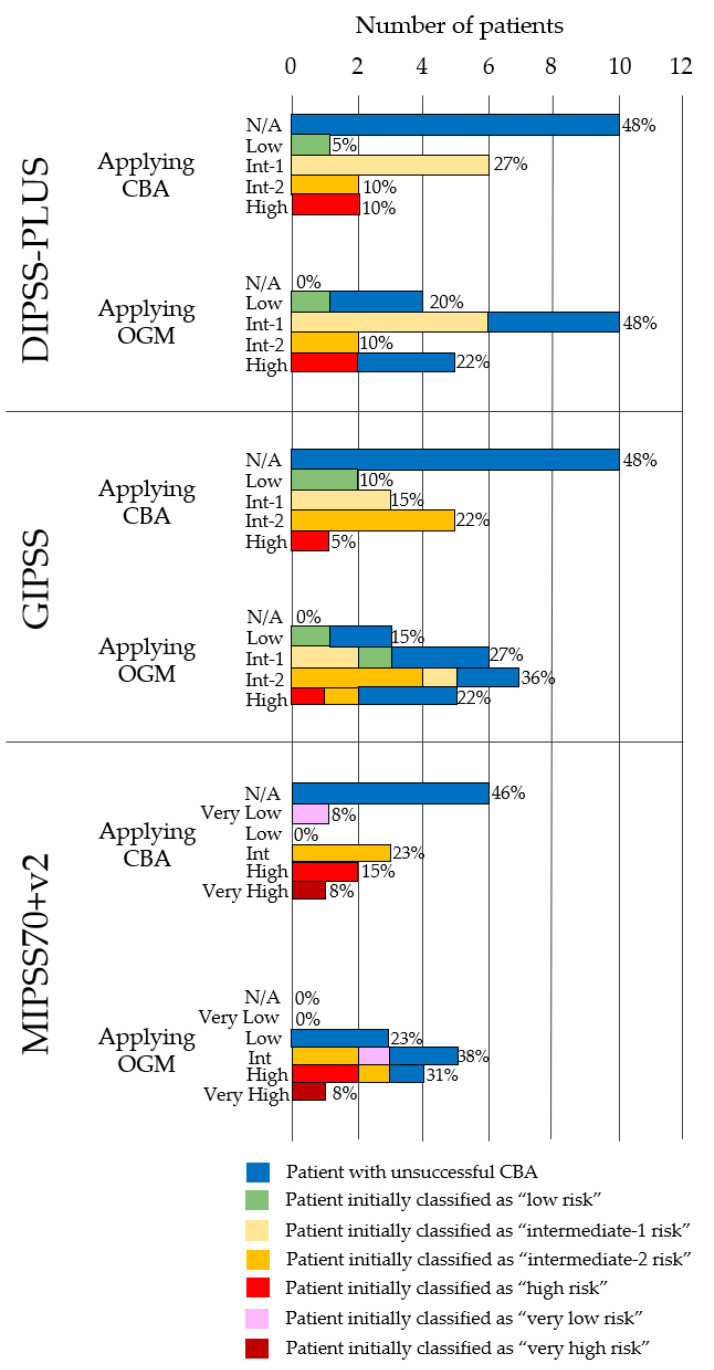
Patient reclassification according to DIPSS-PLUS, GIPSS, and MIPSS70+v2. Applying CBA: calculated with baseline characteristics, NGS, and CBA results. Applying OGM: calculated with baseline characteristics, NGS, and OGM results. All patients who initially could not be classified due to lack of karyotype information were classified via OGM. The increase in the number of patients who upgraded to high-risk categories after reclassification is noteworthy. MIPSS70+v2 was calculated in patients below 70-years-old (*n* = 13). N/A = not applicable due to unsuccessful karyotype.

**Table 1 cancers-15-03039-t001:** Baseline characteristics of the MF cohort (*n* = 21).

Gender	*n* (%)
Men	14 (66%)
Women	7 (34%)
Median age (range)	63 (50–85)
Diagnosis	*n* (%)
PMF	12 (57%)
PPV-MF	2 (10%)
PET-MF	7 (33%)
Disease characteristics	Median (range)
Hemoglobin (g/dL)	10.5 (7.3–14.7)
Leukocytes (×10^6^/µL)	8.17 (3.15–41.01)
Platelets (×10^3^/µL)	268 (17–777)
Circulating blasts (%)	1 (0–8)
Bone marrow fibrosis grade	3 (2–3)
Other characteristics	*n* (%)
Presence of constitutional symptoms	7 (33%)
Transfusion dependency	6 (28%)
Karyotype	*n* (%)
Unsuccessful	8 (38%)
Normal	9 (42%)
Abnormal	4 (20%)
Driver mutation	*n* (%)
*JAK2*	12 (58%)
*CALR*	7 (34%)
*MPL*	1 (4%)
Triple-negative	1 (4%)
Other mutations	*n* (%)
*ASXL1*	9 (43%)
*U2AF1*	3 (14%)
*CBL*	2 (10%)
*EZH2*	2 (10%)
*RUNX1*	2 (10%)
*SRSF2*	2 (10%)
*TET2*	2 (10%)
*TP53*	2 (10%)
*ETV6*	1 (5%)
*PTPN11*	1 (5%)
*SF3B1*	1 (5%)
*ZRSR2*	1 (5%)

**Table 2 cancers-15-03039-t002:** Molecular and cytogenetic results of the cohort.

ID	NGS	CBA	OGM
1	*JAK2* (p.Val617Phe; VAF 13%)	Unsuccessful	ogm[GRCh38] (1–22,X) × 2
2	*JAK2* (p.Val617Phe; VAF 41%), *ASXL1* (c.1720-2A>A; VAF 45%), *SRSF2* (p.Pro95His; VAF 42%) *ETV6* (p.Arg105*; VAF 41%)	Unsuccessful	ogm[GRCh38] (1–22) × 2,(X,Y) × 1
3	*CALR* (p.Leu367Thrfs*?; VAF 39%)	46,XX [18]	ogm[GRCh38] (1–22,X) × 2
4	*CALR* (p.Leu367Thrfs*?; VAF 44%); *ASXL1* (p.Arg1068*;VAF 6.3%)	46,XX,del(X)(q22) [12]/ 46,XX,der(6),t(1;6)(q10;p10) [2] /46,XX [1]	ogm[GRCh38]1q21.2q23.2x3, 6p25.3p22.1x1, t(12;17)(q24.31;p13.1) *KMD2B::TP53* fusion gene Xq11.1q28x1
5	*JAK2* (p.Val617Phe; VAF 50%), *U2AF1* (p.Ser34Tyr; VAF 6%) and *RUNX1* (p.Ala187Thr; VAF 24%)	Unsuccessful	ogm[GRCh38]1q21.1q44x3, 7q11.21q36.3x1, (9)x3
6	*CALR* (p.Leu367Thrfs*?;VAF 31%) and *TET2* (p.Leu1322Pro; VAF 22%)	46,XX [15]	ogm[GRCh38]4q24x1
7	*JAK2* (p.Val617Phe; VAF 17%), *ASXL1* (p.Arg965*; VAF 8.8%), *ASXL1* (p.Gly646Trpfs*12; VAF 30%), *EZH2* (p.Cys565Alafs*110; VAF 33%), *PTPN11* (p.Tyr63Cys; VAF 31%) and *U2AF1* (p.Tyr158_Glu159dup; VAF 41%)	Unsuccessful	ogm[GRCh38] (1–22) × 2,(X,Y) × 1
8	*MPL* (p.Trp515Leu; VAF 43%), *RUNX1* (p.Leu112Val; VAF 39%), *SRSF2* (p.Pro95.Arg102del; VAF 40%) and *TP53* (p.His179Arg; VAF 23%)	46,XY [15]	ogm[GRCh38] (1–22) × 2,(X,Y) × 1
9	*JAK2* (p.Val617Phe; VAF 86%), *ASXL1* (p.(Gly646Trpfs*12; VAF 37%) and *CBL* (p.Lys382Glu; VAF 27%)	46,XY [20]	ogm[GRCh38]9p24.2p13.3x2 hmz
10	*EZH2* (p.R690H; VAF 91%) and *ASXL1* (p.5665Lfs*3; VAF 46%)	47,XY,del(7)(p10), +8 [10]	ogm[GRCh38]t(2;11)(q37.1;q23.2) 7p21.1p11.2x1, t(7;11)(q31.31;q24.1), 7q31.31q32.1x1, (8)x3, 11q23.2q24.1x1
11	*JAK2* (p. Val617Phe; VAF 37%)	46,XY,t(7;13)(q35;q12) [20]	ogm[GRCh38]7q34q35x1, t(7;13)(q34;q14.2), 13q14.13q14.2x1, 20q11.21q11.22x1
12	*CALR* (p.Leu367Thrfs*?; VAF 43%), *CBL* (p.Tyr371His; VAF 3%)	Unsuccessful	ogm[GRCh38] (1–22,X) × 2
13	*CALR* (p.Leu367Thrfs*?; VAF 42%)	Unsuccessful	ogm[GRCh38] (1–22,X) × 2
14	*JAK2* (p.Val617Phe; VAF 80%), JAK2 (p.Cys618Tyr; VAF 80%), SF3B1 (p.Gly740Arg; VAF 45%), *TP53* (p.Ala138Val; VAF 21%) and *ZRSR2* (p.Glu133Glyfs*11, VAF 25%)	46,XY [20]	ogm[GRCh38] (1–22) × 2,(X,Y) × 1
15	*JAK2* (p.Val617Phe; VAF 38%) and *ASXL1* (p.Gly646Trpfs*12; VAF 34%)	Unsuccessful	ogm[GRCh38] (1–22) × 2,(X,Y) × 1
16	*JAK2* (p.V617F; VAF 45%) and *ASXL1* (p.A654Rfs*9; VAF 22%)	46,XY [20]	ogm[GRCh38]2p23.3p23.2x1, 14q32.12q32.31x1, t(2;14)(p23.2;q32.12)
17	*JAK2* (p.Lys539Ile; VAF 48%) and *TET2* (p.Arg1404*; VAF 2.8%)	Unsuccessful	ogm[GRCh38] (1–22,X) × 2
18	*JAK2* (p.Val617Phe; VAF 29%) and *U2AF1* (p.Gln157Pro; VAF 38%)	46,XY [20]	ogm[GRCh38] (1–22) × 2,(X,Y) × 1
19	*JAK2* (p.(Val617Phe; VAF 62%)	47,XY,+9 [1]/46,XY [4]	ogm[GRCh38] (9)x3, 20q11.21q13.32x1
20	*CALR* (p.Lys385Asnfs*?; VAF 40%) and *ASXL1* (p.Gly646Trpfs*12; VAF 33%)	Unsuccessful	ogm[GRCh38] (1–22) × 2,(X,Y) × 1
21	*CALR* (p.L367Tfs*?; VAF 42%)	Unsuccessful	ogm[GRCh38]t(1;12)(p35.2;q13.13), t(1;14)(p35.2;q32.31)

**Table 3 cancers-15-03039-t003:** Patient reclassification via DIPSS-PLUS, GIPSS, and MIPSS70+v2 after OGM.

ID	Applying CBA	Applying OGM
Patient	DIPSS-PLUS	DIPSS-PLUS
	GIPSS	GIPSS
	MIPSS70+v2 *	MIPSS70+v2 *
1	N/A	High
	N/A	Int-1
	-	-
2	N/A	Int-1
	N/A	High
	-	-
3	Int-1	Int-1
	Low	Low
	Int	Int
4	High	High
	Int-1	Int-1
	-	-
5	N/A	High
	N/A	High
	-	-
6	Low	Low
	Low	Int-1
	Very low	Int
7	N/A	High
	N/A	High
	-	-
8	Int-1	Int-1
	Int-2	Int-2
	-	-
9	Int-2	Int-2
	Int-2	Int-2
	-	-
10	High	High
	High	High
	Very high	Very high
11	Int-1	Int-1
	Int-2	Int-2
	High	High
12	N/A	Int-1
	N/A	Low
	N/A	Low
13	N/A	Int-1
	N/A	Low
	N/A	Low
14	Int-2	Int-2
	Int-1	Int-1
	-	-
15	N/A	Low
	N/A	Int-2
	N/A	Int
16	Int-1	Int-1
	Int-2	High
	High	High
17	N/A	Low
	N/A	Int-1
	N/A	Low
18	Int-1	Int-1
	Int-2	Int-2
	Int	Int
19	Int-1	Int-1
	Int-1	Int-2
	Int	High
20	N/A	Int-1
	N/A	Int-2
	N/A	High
21	N/A	Low
	N/A	Int-1
	N/A	Int

Abbreviations: Applying CBA: calculated with baseline characteristics, NGS, and CBA results. Applying OGM: calculated with baseline characteristics, NGS, and OGM results. N/A = not applicable due to unsuccessful karyotype; * = MIPSS70+v2 not calculated because of an age over 70-years-old; Int-1= intermediate 1; Int-2 = intermediate 2; Int = intermediate. In bold are those patients who could be classified using the prognostic scores or who changed their risk category after OGM.

## Data Availability

For inquiries about original data please contact such_esp@gva.es or diaz_alv@gva.es.

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
