# Peer review of "Cytogenetic Assessment and Risk Stratification in Myelofibrosis with Optical Genome Mapping"

_cancers, 2023, doi:10.3390/cancers15113039_

Round 1

Reviewer 1 Report

Diaz-Gonzalez and colleagues present "Optical Genome Mapping improves cytogenetic assessment and risk stratification in myelofibrosis" for consideration in a special issue of the journal cancers.  The premise of the manuscript is to compare Optical Genome Mapping to current standard of care cytogenetic techniques.  It is well known that karyotyping in myelofibrosis samples is challenging and has a high failure rate.  Karyotypic rearrangements are a reliable stratification tool (e.g. DIPSS) for treatment however, the high failure rate by CBA leaves many patients without appropriate information for stratification.  As such, the investigation of the use of Optical Genome Mapping as an alternative to karyotyping is timely.   

The manuscript is well written and compares a reasonable sized cohort for a proof of concept investigational study between standard of care cytogenetics versus OGM.  

This reviewer would like to make the following comments for revision in the manuscript.  

Point 1 - Methods

line 163  "Bionano’s human control sample SV database which contains variants collected from ethnically-diverse mapped human genomes with no reported disease phenotypes. Ultimately, a manual inspection of each alteration by at  least two reviewers was performed. All clinically significant alterations were reported." 

The authors should define if they excluded all SVs in the population database or only those above a certain percentage.  What was the filter setting for the populational database...0%, 1%, etc.?  

Also, did the authors exclude SVs that were in masked regions?  In the filter settings you can select whether to show only Non-masked SVs and copy number abnormalities or ALL structural variants and copy number variants (excluding those removed by the normal population).  Please clarify which setting.  

Also, what were the criteria used to determine if a variant was clinically significant?  Did it need to be above a certain size, have a VAF >5%, have at least 10 molecules.  

Point 2 - True versus significant calls
Line 195  "Manual inspection was performed to select significant chromosomal alterations. In total, 8 SVs, 17 CNVs and 1 CN-LOH were considered true calls."

Do the authors mean "true" or "clinically significant".  It is likely for a call to be "true" but have no clinical significance (e.g. Tier 3 or 4) and therefore not be reported/included in the analysis.  Please clarify this statement so the reader understands if this step was to remove "artefacts" or "significant" calls.  And if any criteria were used.  

Point 3:  Comparability of Bone Marrow and Peripheral Blood

Was any analysis performed on blood and bone marrow concurrently (CBA and OGM)?

Separately, the authors are also internally inconsistent in the manuscript regarding the comparability of PB and BMA specimens. 

For example, in line 289 "The low level of circulating disease in MF normally prevents to perform reliable CBA from PB samples."

However, a few lines later on, line 300,  "Thus, performing a cytogenetic study by OGM in PB is a suitable alternative in these patients.", contradicts that said in line 289.  It also is contradicted by previously published literature and the author's own results.

This reviewer would also like to point out that the limit of detection for CBA and OGM is comparable.  So pointing to OGM as having a larger dynamic range than karyotype would not be accurate.  A standard karyotype analysis of 20 cells has the 'statistical power to exclude mosaicism" of approximately 10-14% depending on confidence (Hook, 1977).  This is at the cell level and translates to a "VAF" of 5-7%.  While this is a very rough calculation that doesn't take into account many variables (e.g. technologist bias, chromosome quality etc) it does suggest that the limit of detection for OGM and karyotype is similar.  This reviewer would suggest altering this point as it contradicts the main thesis (and data) of your manuscript.   

Reference showing comparability of cyotgenetics in peripheral blood and marrow:  Tefferi, A., Meyer, R. G., Wyatt, W. A., & Dewald, G. W. (2001). Comparison of peripheral blood interphase cytogenetics with bone marrow karyotype analysis in myelofibrosis with myeloid metaplasia. British Journal of Haematology, 115(2), 316–319. https://doi.org/10.1046/j.1365-2141.2001.03131.x

Point 4:  Functional consequences of novel translocations

Line 312:  :In our series, we disclosed one potential KDM2B::TP53 fusion gene in patient #4 that may be an epigenetic factor for leukemogenesis, as increased KMD2B expression facilitated the proliferation of hemato- poietic progenitor cells and induced leukemic transformation [27]"...

Line 321 "Functional studies should be performed to elucidate the role of this fusion gene."

Discussion of KDM2B::TP53 fusion should likely focus more on the more likely possibility that this translocation may simply just disrupt TP53 function (LOF).  Without supporting RNA data that a functional transcript exists the authors should likely be more circumspect about this being a "fusion" and rather refer to it as a potentially disruptive translocation.  

Point 5:  Description of cryptic events.  This section in results was a bit confusing and there many events that could be given a bit more detail to enhance their significance and clarity for the reader.  

Line 218 "OGM also detected additional cryptic aberrations not identified with standard techniques in 7 out of 11 patients with informative karyotypes. Cases #6, #11 and #19 presented cyto-genetically missed deletions. Patient #6 displayed a 376 kbp deletion on the long arm of chromosome 4 (region 4q24), originating a complete deletion of TET2.

--would suggest "Patient #6 had a 376 kb deletion on the long-arm of chromosome 4 (4q24), resulting in a deletion of one copy of TET2".  ...a "complete deletion" may suggest a homozygous deletion.  

Line 221 Deletions on chromosome 20 were uncovered both in patient #11 (20q11.21q11.22) and #19  (20q11.21q13.32).  

How big were they?  Were they submicroscopic?  Was ASXL1 also deleted?  

Line 223 - "Furthermore, OGM disclosed complex events like the translocations t(2;11)(q37.1;q23.2) and t(7;11)(q31.31;q24.1) with material loss of adjacent regions on patient #10 and the translocation t(2;14)(p23.2;q32.12) with deletion of adjacent regions in patient #16. 

The word "complex" is very loaded for a clinical audience.  It should be used carefully.  It can be easily confused with 'karyotypic complexity' - which is not the case here.  Consider, "OGM showed additional rearrangement complexity in patients with t(2;11) and t(7;11) with deletions flanking the translocation breakpoints".  

Point 6 - De novo analysis

line 234 "De novo analysis did not detect additional SVs previously identified by RVP." 

There is very little information given on the de novo analysis in the manuscript.  The de novo analysis is known to have a lower limit of detection of approx 25% VAF for events.  Perhaps a supplemental table showing the comparison of results between the de novo and RVP. I also found it somewhat surprising that the de novo analysis only detected LOH of 9p in one sample.  This is often a frequent finding in MPN and especially MF.  Only patient 9 is reported to have a hmz region on 9.  In any event, a table showing if de novo missed any calls that RVP detected would be informative.

Point 7: 
Line 293 - "On the other hand, to perform a multi-probe FISH panel covering all the SVs with prognostic impact on MF is a laborious, time-consuming and expensive strategy, not feasible in routine clinical practice.

Not only is FISH profiling considered time consuming and expensive, it is not considered sufficient to properly classify patients and karyotyping is recommended as the gold standard.   The revised cytogenetics classification published by Tefferi et al 2018 requires knowledge about whether an abnormality is a sole abnormality for appropriate classification - this is not information that can be discerned from FISH analysis alone.  

Arber, D. A., Orazi, A., Hasserjian, R. P., Borowitz, M. J., Calvo, K. R., Kvasnicka, H. M., Wang, S. A., Bagg, A., Barbui, T., Branford, S., Bueso-Ramos, C. E., Cortes, J. E., Dal Cin, P., DiNardo, C. D., Dombret, H., Duncavage, E. J., Ebert, B. L., Estey, E. H., Facchetti, F., … Tefferi, A. (2022). International Consensus Classification of Myeloid Neoplasms and Acute Leukemias: integrating morphologic, clinical, and genomic data. Blood, 140(11), 1200–1228. https://doi.org/10.1182/blood.2022015850 

Point 8 - Dicentric chromosomes and other OGM weaknesses

Line 342 - "Therefore, this technique is not able to detect dicentric chromosomes and whole-arm translocations such as Robertsonian translocations. "

Technically, OGM can detect dicentric chromosomes.  While it doesn't show the individual structure of each chromosome, as in a karyotype, it does capture the rearrangements (dicentric chromosome breakpoints often occur in euchromatic sequence).  OGM also can't see the structure of an isochromosome - although the imbalance in copy number is evident. 

Consider changing the comment about not detecting dicentric chromosomes to be clearer about what OGM can see and can't see.

Point 9:  Discordance between FISH & CBA vs OGM & Array

Line 348.  "Perhaps, CBA and FISH could have missed this translocation due to a growth advantage of the 46,XY cultured cells over the t(2;14) cells."

Which FISH was performed?  Was is interphase FISH or sequential G to FISH.  Table S2 provides no additional information.  Only sequential G to FISH would suffer from metaphase bias, interphase FISH should have similar frequency to OGM.  I think a more likely explanation of the translocation being missed by karyotyping is that both breakpoints are very distal.  It may indeed be a cryptic rearrangement considering that CBA for MF generally results in a chromosomes with a band level resolution of 300-350.  It would be easy to imagine how it could be missed by CBA at that resolution.  Loss of the entire 2p distal to 2p23.2 is approx 28MB of sequence.  If replaced with ~12MB of chr14 the total displaced sequence would be 16 MB - at the limit of detection for G-banding at 350 bands.  This reviewer would suggest modifying this point to expand upon the potential causes of discrepancy and perhaps even including a supplemental figure showing CBA, FISH and OGM data in case the reader cared to see more detail.  I also wonder if the rearrangement may actually be a cryptic insertion flanked by deletions.  If the breakpoints flank the deletions it suggests the material is inserted rather than translocated and this would provide a more rational explanation for the discordance between results.  It may be easier to see the mechanism of rearrangement with the de novo analysis in this case. 

Point 10:  Tables and Figures

Table S2 - as per point above, Table S2 is lacking information about which FISH probes were used.  Also, microarray nomenclature could be given for copy number alterations and would strengthen the rigor of this information for readers.  

Figure 2.  Data labels to show percents for each bar in the histogram in each group, or perhaps a data table below the figure would allow the reader to easily see the numeric value of reclassification the author's are showing with this figure.  E.g. in before OGM DIPSS plus ~50% of the patients were N/A.  Showing how this is redistributed among the groups would make a more enticing figure.  Alternatively, data series could be used showing where cases went using a hatched or alternatively coloured bar on top of the original histogram data showing the distribution of cases added to each group or re-classified.  

Reviewer 2 Report

Authors report on application of OGM in 21 MPS patients.

major comments:

- authors need to discuss the fact that MPS-cells are present in mosaic with normal, non-cancer cells. Thus, OGM may miss aberrant clones below a certain cut-off. 

- state what is thecut-off rate of OGM. is it 5, 10, 20, 30%?

- state that OGM did not detect any genetic chnages in 12/21 cases. Accordingly, OGM was not helpful  in >50% of the cases. Please revise title, abstract and overall conclusions, which are misleading by now. OGM combined with NGS and karyotype may lead to 100% of risk stratifications - but the ~60% of normal result cases in OGM were only informtive as NGS found some aberrations. 

minor comments:

- Check name of Ref 24 in line 24 - it is since 2 editions no longer 'cytogenetic' but 'cytogenomic'

- replace 'conventional cytogenetics' by 'banding cytogenetics'

Round 2

Reviewer 2 Report

Thanks for amendments in the paper

Still in results it is given  the impression that there is a difference in classifying cases when only doing GTG-banding and when only doing OGM.

This is incorrect - please clrealy state in text and Table and figures that you compare pure banding cytogenetic results with results after NGS plus OGM.

Also make clear which / how many cases really profited from additonal NGS only and additional OGM only - as far as this referee sees most progress in classifying comes due to NGS and not due to OGM. 

Author Response

Kind regards

Round 3

Reviewer 2 Report

Authors still have not reviesed the text accordingly - they insist to reptort big success of using OGM additionally to NGS and banding cytogenetics  see line 95 and 410 following.